# The optimal warming strategy to reduce perioperative hypothermia: A prospective randomized non-blinded clinical trial

Ronak Desai[1,2,3]*, Jason Gosschalk[1,2], Noud van Helmond[1,2,3], Ludmil Mitrev[1,2,3], Catherine Zhang[1,2], Brian McEniry[2,3], Krystal Hunter[2,4], Ernest Wallace[1,2], Michele Mele[1,2,3], Jennifer Ocbo[1,2,3], Keyur Trivedi[1,2,3], George Hsu[1,2,3], Sandeep Krishnan[5], John Dibato[2,4], Kinjal Patel[1,2,3]

1 Cooper Medical School of Rowan University, Camden, New Jersey, United States of America, 2 Cooper University Hospital, Camden, New Jersey, United States of America, 3 Department of Anesthesiology, Cooper University Hospital, Camden, New Jersey, United States of America, 4 Cooper Research Institute, Cooper University Hospital, Camden, New Jersey, United States of America, 5 Department of Anesthesiology, Wayne State University School of Medicine, Pontiac, Michigan, United States of America

* Desair@rowan.edu

## Abstract

### Background

To mitigate perioperative hypothermia, patients can be warmed preoperatively and intraoperatively with forced-air warming (FAW) and conductive warming (CW) methods. We examined the association of four combinations of pre- and intraoperative CW and FAW with the magnitude of intraoperative hypothermia.

### Methods

We conducted a prospective randomized trial at a tertiary healthcare center in the United States (trial registration number ISRCTN23065394). Patients were randomized to 4 arms based on the following pre/intraoperative warming combinations: (1) CW/CW, (2) FAW/FAW, (3) no active prewarming (NAPW)/CW, (4) NAPW/FAW. Body temperature was measured using an esophageal probe. The area under the temperature curve (AUC) below 36°C was calculated according to the trapezoidal rule and quantified intraoperative hypothermia. A mixed model was used to estimate differences in AUC between the 4 arms.

### Results

182 patients were analyzed. Patients in the NAPW/FAW arm had the highest AUC values while those in the CW/CW arm had the lowest. AUC values [median (Q1, Q3] were as follows: CW/CW = 4.7 (0, 26.6); FAW/FAW = 8.0 (0, 30.8); NAPW/CW = 7.4 (0, 27.1); NAPW/FAW = 19.9 (5.0, 44.3). Mixed model results showed significant lower AUC values in CW/CW and NAPW/CW when compared to NAPW/FAW. The ratio

**Data availability statement:** The data set is accessible through figshare: https://figshare.com/articles/dataset/study_dataset_for_The_Optimal_Warming_Strategy_to_Reduce_Perioperative_Hypothermia_A_Prospective_Randomized_Non-Blinded_Clinical_Trial_by_Desai_Ronak_DO_et_al_/25377418/1

**Funding:** The HotDog™ warming system was provided by HotDog™, owned by Augustine Medical Inc. This study was also supported by an unrestricted grant from Augustine Medical Inc. The funder had no role in the study design, data collection, interpretation, or drafting of the manuscript.

**Competing interests:** The authors have declared that no competing interests exist.

of mean AUC [95% CI] between CW/CW vs NAPW/FAW was 0.49 [0.24, 0.98], 51% lower, and between NAPW/CW and NAPW/FAW, 0.46 [0.23, 0.91], 54% lower. When the AUC was normalized to the duration of surgery (AUC/case duration in°C, or "relative AUC"), significant lower relative AUC values were observed between FAW/FAW vs NAPW/FAW (48% lower, p = 0.0419) and NAPW/CW vs NAPW/FAW (48% lower, p = 0.0407).

## Conclusions

CW is more effective than FAW at reducing intraoperative hypothermia when FAW is used without prewarming. When patients are actively prewarmed, CW and FAW show no difference in their ability to maintain patient temperature.

## Introduction

Perioperative hypothermia (PH) is a common surgical complication, defined as core body temperature below 36°C [1]. PH is associated with various adverse outcomes, including increased blood loss, increased time to emerge from anesthesia, prolonged hospitalization, surgical site infections, and patient discomfort [2–6]. Several interventions assist with patient temperature maintenance: adjusting ambient room temperature, prewarming patients, and applying warming devices intraoperatively. There is no consensus regarding the benefit of prewarming (in addition to intraoperative warming), although a large Cochrane library review did find one study that supported superiority of prewarming in patients undergoing major abdominal surgery [7]. The same review concluded there is no superiority of one active surface body warming system to another [7].

Two common methods of surface warming have been referred to as "forced-air" and "resistive" warming. High-resistance circuits produce heat in both methods. Forced-air warming (FAW) devices apply convective heat, warming air within a high-resistance circuit which then blows over the patient. FAW generally utilizes perforated plastic sheets ducted to a heating tower that produces warm, filtered air. There are concerns that FAW may disrupt laminar flow and subsequently contaminate the surgical field, though it is unlikely to increase infection incidence [8–10]. Recently, conductive warming (CW) devices have gained popularity due to their ability to directly transfer heat to the patient. CW bypasses air as an intermediate, which improves heat transfer compared to FAW [11]. While CW devices can vary in efficiency, they may offer more efficient heat transfer than FAW [12].

Studies comparing CW and FAW efficacy have yielded mixed results; several have determined that CW is non-inferior to FAW [13–17]. Alternatively, one study concluded that FAW was superior to CW with under-body mattress alone (without blanket) [18] and another study concluded that FAW was superior to CW with lower-body blanket alone (without under-body mattress) [19]. In a meta-analysis of five studies, FAW was found to be superior to CW in preventing hypothermia [20].

Prewarming may also lessen PH. Several studies have noted a risk-reduction of PH with prewarming, suggesting that ten or more minutes may be protective [21,22]. Similarly, prewarming may reduce surgical site infection rate [23].

The aim of this study was to compare the efficacy of four combinations of FAW and CW pre- and intra-operatively. Our main hypothesis was that there would be a significant difference in the mean area under the curve (AUC) of temperature below 36°C during the intraoperative period between the four combinations of warming strategies. A secondary goal was to examine whether active prewarming was associated with less intraoperative hypothermia.

## Materials and methods

### Participants

Patients scheduled for surgery were screened for eligibility. Inclusion criteria were elective surgery under general anesthesia projected to last greater than 90 minutes but less than 240 minutes. The expected duration of 90–240 minutes was selected because, based on clinical experience, it would allow for any hypothermia to occur while limiting the influence of any substantial outliers in very long surgical cases. Exclusion criteria were cardiac or vascular surgery, pregnancy, age under 18 years old, incarceration, inability to provide written informed consent, or not being fluent in English. The orthopedic surgeons in our institution only use CW as the standard of care and therefore their patients were not available for inclusion in this study.

### Ethics approval and registration

Ethical review and oversight was provided by the Cooper University Health Care Institutional Review Board (study number 18–148). All participants provided signed informed consent before participation in the study. The trial was registered retrospectively in the ISRCTN registry, registration number ISRCTN23065394 (https://doi.org/10.1186/ISRCTN23065394). At the time of trial commencement, it was not mandatory for publication to register trials prospectively, unless funded by public sources such as the National Institutes of Health. Furthermore, although this work meets the criteria for a clinical trial, it involves the study of established, non-invasive, approved methods of patient warming. A combination of some or all of these methods would have been used even if the subjects were not in the trial, as it is standard of care to warm patients perioperatively. Lastly, this was a single-institution, investigator-initiated trial that did not advertise for enrollments. Those were the reasons for not registering the trial prospectively. The study recruitment period was from October 3rd, 2019 (initial IRB approval notice date) to July 8th, 2022 (date of final enrollment). Actual enrollments began on 12/4/2019 and ended 7/8/2022. This manuscript adheres to the Consolidated Standards of Reporting Trials (CONSORT) guidelines.

### Warming and temperature monitoring procedure

The FAW warming system (Bair Hugger™, 3M, Maplewood, MN) was set to 43°C and was attached to either an upper or lower body blanket, whichever was amenable to intraoperative surgical needs. Patients prewarmed with FAW were given lower body FAW blankets. The CW (HotDog™, Augustine Medical, Eden Prairie, MN) under-body mattress was set to 39°C, and the CW blanket was set to 43°C. CW blankets were either upper or lower body, depending on surgical need. A shortened under-body mattress (WaffleGrip™, Augustine Medical) was used to accommodate surgeries requiring lithotomy or Trendelenburg positioning. Preoperatively, CW patients were warmed with a lower body blanket and without an under-body mattress. The temperature settings used reflect the maximum temperature setting of each device, to standardize each intervention to its highest-output mode. Consistent with a previous randomized controlled trial on warming strategies [24], patient temperature was measured (to the tenth degree Celsius) every 15 minutes intraoperatively using an esophageal probe placed after induction of anesthesia. IV fluids were not warmed. Room temperature was measured every 15 minutes using an Elitech RC-5 temperature logger (Elitech Technology Inc., San Jose, CA). Patients were contacted twice postoperatively, at 24 hours and then 30 days to assess for any complications not recorded in the electronic medical record.

## Study variables

**Primary outcome.** To simultaneously assess the duration and degree of hypothermia under 36°C during surgery, the AUC under 36°C was calculated for each patient according to the trapezoidal rule for calculating definite integrals between each 15-minute time point (AUC = 0.5*[$T_1$-36°C + $T_2$-36]*[15 minutes]) where $T_1$ = starting core body temperature and $T_2$ = the core body temperature 15 minutes later, calculated for the duration of the surgery [25]. To prevent over- or under-estimation of hypothermia when patients' temperatures cross 36°C, we first assumed a linear temperature change between each 15-minute measurement. We then calculated the x-intercept ($x_{y=36°C}$) when temperature crosses 36°C, using this value as the trapezoidal base when calculating AUC (0.5*[$T_1$-36°C + $T_2$-36][$x_{y=36°C}$]). AUC calculation for a representative patient is illustrated in Fig 1.

**Secondary outcomes.** Relative case hypothermia (AUC/case duration, in °C) was calculated by dividing the AUC by the duration of the surgery, up to 240 minutes. Incidence of hypothermia was defined as any single core temperature measurement below 36°C intraoperatively.

**Treatment group.** We defined four arms based on the pre-warming/intraoperative warming interventions as follows: (1) CW/CW, (2) FAW/FAW, (3) no active prewarming (NAPW)/CW, (4) NAPW/FAW

## Study design

**Randomization.** To minimize natural variation among the participants from different surgical specialties and to broaden applicability across surgery types, a randomized blocked design was carried out with surgery type being the blocking factor. Ten surgery units were formed via a combination of procedure type (abdominal, gynecologic, breast, plastic/reconstructive, and urological surgery) and expected case length (greater or less than 150 minutes). The participants were ultimately randomized to the 4 arms within the 10 surgery units using a block randomization table created using the random number generator function in Microsoft Excel (S1 Appendix). The randomization table was created before the first patient was screened into the study by one of the investigators (NVH). Only the study coordinator had access to the randomization table thereafter. Patients who fit the pre-defined randomization units were approached sequentially as they arrived for surgery and randomized according to the table if they provided written informed consent to participate in the study on the day of surgery.

**Sample size and power analysis.** Sample size estimation was based on a null hypothesis of equal means among the 4 treatment arms. An AUC of 1.0°C*hour difference between the treatment arms is considered a clinically relevant difference that manifests for a subject remaining 1.0°C below 36°C for an hour. Prior studies have shown hypothermia to occur in as many as 45% of patients after 100 minutes post induction with only intraoperative active warming [26]. Using the R functions *Fpower1* and *Fpower* from the R package *daewr* [27] with a standard deviation in the AUC < 36°C of 0.3 and assuming similar variability across the treatment arms [16, 25, 28], a minimum of 216 participants (6 replicates x 4 arms x 9 blocks) with 54 per arm, and power of approximately 80% would be sufficient to detect a 1.0°C*hour difference in AUC < 36°C between the 4 arms.

## Statistical analysis

Data values on patient's characteristics at baseline were summarized separately by the four treatment arms as count (%) for categorical variables and mean (SD) or median (Q1, Q3) for continuous variables. Boxplots were used to illustrate the distributions of the outcomes between the 4 rewarming arms. After adding a constant of 1, the outcomes were log transformed and normality assessed using Quantile-Quantile plots. A mixed model was used to estimate the differences in the outcomes between the 4 arms while accounting for variability between the 10 surgery units. In this model the 4 arms were treated as fixed effect variable while the 10 surgery units as random effect variable. The differences in the model estimates between the arms are expressed as relative mean (95% CI). Significant differences between arms are realized when the 95% CI does not contain 1. A full description of the model and the effects of interest is provided in S2 Appendix.

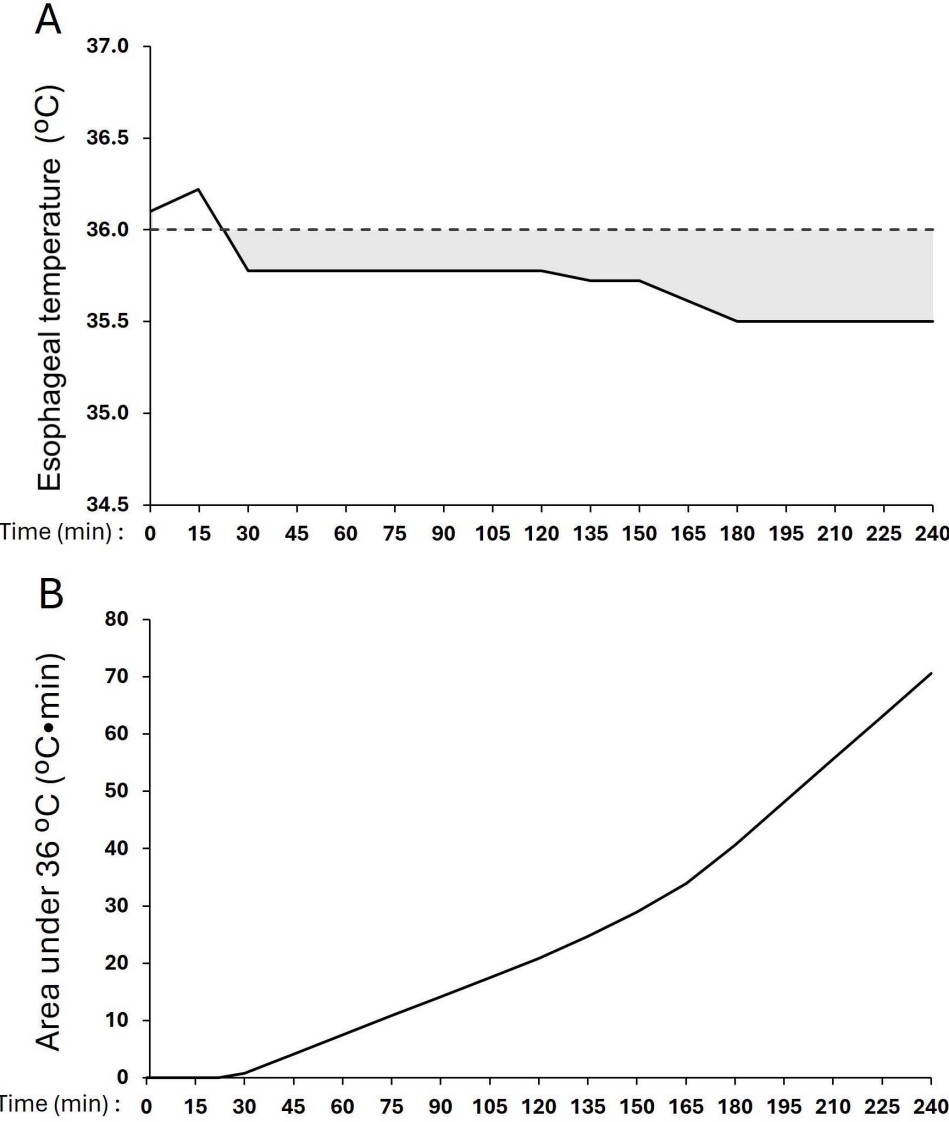

**Fig 1. Area under the curve (AUC) under 36°C for a representative patient.** Panel **a** depicts body temperature during surgery with the shaded area reflecting the AUC under 36°C. Panel **b** shows the corresponding AUC in°C*min.

Secondary safety and outcome differences for length of stay, estimated blood loss, and patient comfort (Likert scale 0–10) were analyzed using Kurskal Wallis test. Secondary analyses for complications as binary outcomes (unexpected bleeding, transfusion, any other immediate complications, readmission and any post-operative complication noted at 30 days) were assessed using Chi square or Fisher's exact test. All analyses were done using SAS 9.4 (SAS Institute, Inc., Cary, NC) and R 4.2.2, and conclusions were made at a 5% significance level.

## Results

A total of 184 participants were enrolled. Two subjects were removed before data collection due to surgical team preference for a certain warming technique, leaving a final cohort size of 182 participants (Fig 2).

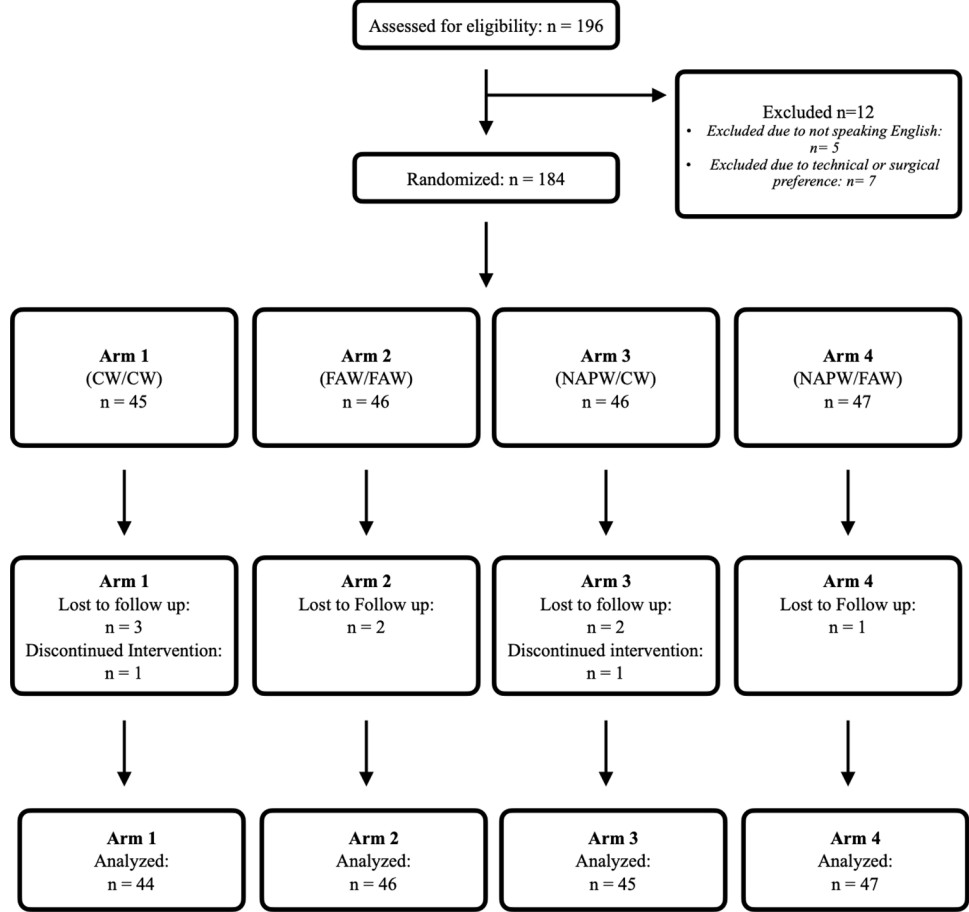

**Fig 2. Consolidated Standards of Reporting Trials (CONSORT) enrollment diagram.** CW, conductive warming; FAW, forced-air warming; NAPW, no active prewarming.

Of those, 44 (24%), 46 (25%), 45 (25%) and 47 (26%) were assigned to the CW/CW, FAW/FAW, NAPW/CW and NAPW/FAW arms, respectively. The mean (SD) age was 49 [16] years with 20% males, mean (SD) BMI of 33.3 (11.6) kg/m², and length of stay of 1.7 days (3.5) (Table 1). Demographic and clinical characteristics were similar across the arms at baseline (Table 1).

Patients in the NAPW/FAW arm had the highest AUC values while those in the CW/CW had the lowest. The AUC values in the other two arms are closer to each other [median (Q1, Q3) AUC: CW/CW = 4.7 (0, 26.6); FAW/FAW = 8.0 (0, 30.8); NAPW/CW = 7.4 (0, 27.1); NAPW/FAW = 19.9 (5.0, 44.3)}. A similar pattern was observed for the relative AUC values between the arms (Table 2, Fig 3).

Table 3 shows the estimated average AUC and relative AUC values for each arm, with NAPW/FAW having consistently higher values while the other arms had lower and similar estimates. Mixed model results showed significant lower AUC values in CW/CW and NAPW/CW when compared to NAPW/FAW. The ratio of mean AUC [95% CI] between CW/CW vs NAPW/FAW: 0.49 [0.24, 0.98], ~51% lower, and between NAPW/CW and NAPW/FAW: 0.46 [0.23, 0.91], ~54% lower. As for the secondary outcomes, significant lower relative AUC values were observed between FAW/FAW vs NAPW/FAW (~48% lower, p = 0.0419) and NAPW/CW vs NAPW/FAW (~48% lower, p = 0.0407). No significant difference was observed in all the other pairwise comparisons (Fig 4).

**Table 1. Distribution of patients' demographics, baseline clinical characteristics, and surgical approach and duration by study arm.**

| Baseline characteristics | Overall | CW/CW | FAW/FAW | NAPW/CW | NAPW/FAW |
|---|---|---|---|---|---|
| Sample size, n | 182 | 44 | 46 | 45 | 47 |
| Age in years, mean (SD) | 48.9 (16.4) | 53.9 (14.8) | 48.5 (17.2) | 46.2 (18.0) | 47.4 (14.7) |
| Male, n (%) | 37 (20) | 12 (27) | 8 (17) | 10 (22) | 7 (15) |
| BMI in kg/m2, mean (SD) | 33.3 (11.6) | 32.0 (7.6) | 34.6 (17.5) | 33.2 (9.3) | 33.5 (9.7) |
| Comorbidity, n (%) | | | | | |
| Hypertension | 64 (35) | 18 (41) | 13 (28) | 16 (36) | 17 (36) |
| Diabetes | 27 (15) | 6 (14) | 10 (22) | 6 (13) | 5 (11) |
| Cancer | 53 (29) | 15 (34) | 15 (33) | 7 (16) | 16 (34) |
| Coronary artery disease | 5 (3) | 3 (7) | 1 (2) | 1 (2) | 0 (0) |
| Peripheral vascular disease | 9 (5) | 3 (7) | 3 (7) | 2 (4) | 1 (2) |
| Cerebrovascular accident | 7 (4) | 4 (9) | 1 (2) | 1 (2) | 1 (2) |
| Chronic kidney disease | 30 (16) | 9 (20) | 7 (15) | 6 (13) | 8 (17) |
| Hypothyroidism | 17 (9) | 3 (7) | 6 (13) | 2 (4) | 6 (13) |
| Hyperthyroidism | 1 (1) | 0 (0) | 0 (0) | 0 (0) | 1 (2) |
| Asthma or COPD | 45 (25) | 15 (34) | 12 (26) | 12 (27) | 6 (13) |
| Liver cirrhosis | 22 (12) | 3 (7) | 7 (15) | 7 (16) | 5 (11) |
| Peripheral neuropathy | 2 (1) | 2 (5) | 0 (0) | 0 (0) | 0 (0) |
| Adrenal insufficiency | 1 (1) | 0 (0) | 0 (0) | 1 (2) | 0 (0) |
| Surgical approach, n (%) | | | | | |
| Open surgery | 83 (46) | 27 (61) | 22 (48) | 17 (38) | 17 (36) |
| Laparoscopy | 99 (54) | 17 (39) | 24 (52) | 28 (62) | 30 (64) |
| Surgery duration in minutes, mean (SD) | 147 (59) | 136 (58) | 156 (61) | 149 (61) | 148 (54) |

BMI, body mass index; COPD, chronic obstructive pulmonary disease; SD, standard deviation; CW, conductive warming; FAW, forced-air warming; NAPW, no active prewarming

**Table 2. Outcomes by study arm.**

| Outcomes | Overall | CW/CW | FAW/FAW | NAPW/CW | NAPW/FAW |
|---|---|---|---|---|---|
| AUC | | | | | |
| mean (SD) | 28.7 (51.7) | 23.0 (42.1) | 29.7 (62.9) | 25.5 (47.5) | 36.4 (52.1) |
| median (Q1, Q3) | 10.3 (0, 32.6) | 4.7 (0, 26.6) | 8.0 (0, 30.8) | 7.4 (0. 27.1) | 19.9 (5.0, 44.3) |
| Relative Hypothermia (AUC/ Procedure length *100) | | | | | |
| mean (SD) | 19.4 (34.1) | 20.1 (44.3) | 17.8 (38.7) | 16.0 (23.9) | 23.4 (26.4) |
| median (Q1, Q3) | 8.1 (0, 23.6) | 5.9 (0, 22.6) | 4.3 (0, 15.5) | 8.0 (0, 16.5) | 11.2 (2.4, 42.5) |
| Operating room temperature | | | | | |
| mean (SD) | 21.3 (0.9) | 21.6 (0.9) | 21.2 (0.9) | 21.3 (0.8) | 21.3 (0.9) |
| median (Q1, Q3) | 21.3 (20.7, 22.0) | 21.4 (20.9, 22.3) | 21.2 (20.5, 21.8) | 21.2 (20.9, 21.8) | 21.3 (20.6, 22.1) |
| Incidence of hypothermia | 134 (74) | 31 (70) | 33 (72) | 30 (67) | 40 (85) |
| Hospital length of stay, days | 1.7 (3.5) | 2.3 (5.4) | 1.5 (2.4) | 1.6 (2.9) | 1.4 (2.5) |

Data values in mean (SD), median (Q1,Q3), or count (%).

SD, standard deviation; CW, conductive warming; FAW, forced-air warming; NAPW, no active prewarming; AUC, area under the curve.

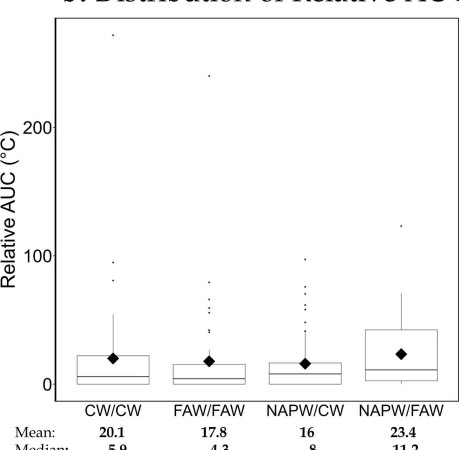

**a. Distribution of AUC**

|        | CW/CW | FAW/FAW | NAPW/CW | NAPW/FAW |
|--------|-------|---------|---------|----------|
| Mean:  | 23    | 29.7    | 25.5    | 36.4     |
| Median:| 4.7   | 8       | 7.4     | 19.9     |

**b. Distribution of Relative AUC**

|        | CW/CW | FAW/FAW | NAPW/CW | NAPW/FAW |
|--------|-------|---------|---------|----------|
| Mean:  | 20.1  | 17.8    | 16      | 23.4     |
| Median:| 5.9   | 4.3     | 8       | 11.2     |

**Fig 3. Box plots of the hypothermic area under the curve, AUC (a) and Relative hypothermic AUC (b) by study arm.** The diamonds within the box represent the mean while the horizontal lines represent the median as illustrated below each study arm.

**Table 3. Mean (95% CI) estimates of the primary (AUC) and secondary (Relative hypothermia) outcomes from the Mixed model for each trial arm.**

| Treatment arms | Mean AUC (95% CI) | Mean Relative Hypothermia (95% CI) |
|----------------|-------------------|-------------------------------------|
| CW/CW          | 6.6 (3.5, 11.9)   | 5.3 (3.0, 9.0)                      |
| FAW/FAW        | 7.1 (3.8, 12.6)   | 4.8 (2.7, 8.1)                      |
| NAPW/CW        | 6.1 (3.2, 11.0)   | 4.8 (2.7, 8.1)                      |
| NAPW/FAW       | 14.6 (8.4, 24.9)  | 10.2 (6.2, 16.5)                    |

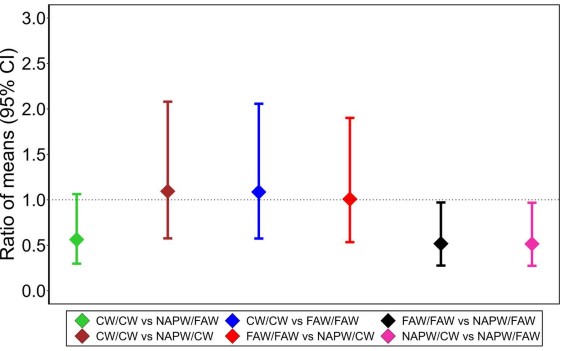

**a. Pairwise comparisons of AUC**

**b. Pairwise comparisons of Relative AUC**

Legend:
- CW/CW vs NAPW/FAW
- CW/CW vs NAPW/CW
- CW/CW vs FAW/FAW
- FAW/FAW vs NAPW/CW
- FAW/FAW vs NAPW/FAW
- NAPW/CW vs NAPW/FAW

**Fig 4. Ratio of mean AUC (95% CI) and relative AUC (95% CI) comparing the primary (a) and secondary (b) outcomes among the 4 treatment arms.** The ratios are estimates from the differences in the log-transformed outcomes of the mixed model. The horizontal dashed lines represent a mean ratio of 1, i.e., no difference between arms.

There were no statistically significant differences between the arms in terms of secondary outcomes or complications. None of the patients had post-operative arrhythmias or heart attacks, and the estimated blood loss, instances of transfusion or self-reported comfort post-operatively did not appear to differ between the arms. The secondary outcomes and complications are shown in Supplementary Appendix 3 in table format.

## Discussion

In this study, we compared the effectiveness of four warming strategies by measuring the degree of intraoperative hypothermia experienced by surgical patients in each arm: CW/CW, NAPW/CW, FAW/FAW and NAPW/FAW. Our findings demonstrated that CW applied with or without active prewarming was associated with significantly less hypothermia than FAW without active prewarming. CW/CW resulted in 51% less hypothermia compared to NAPW/FAW, and NAPW/CW was associated with 54% less hypothermia than NAPW/FAW, as expressed by the mean AUC below 36°C. When expressed as relative AUC (i.e., AUC divided by case duration), FAW/FAW had 48% less hypothermia than NAPW/FAW, and NAPW/CW had ~48% less hypothermia than NAPW/FAW. No significant difference was observed in the pairwise comparison between CW/CW and FAW/FAW, or between NAPW/CW and FAW/FAW.

These findings suggest that CW performed better, both with and without prewarming, than FAW without prewarming, but similarly when compared to FAW with FAW prewarming. FAW prewarming with FAW intraoperative warming was associated with less hypothermia than FAW intraoperative warming alone. No significant difference was found between CW/CW and NAPW/CW. This suggests that the benefit of prewarming may be dependent on the intraoperative warming strategy and thus precludes us from making a blanket statement about the benefits of prewarming in all settings. One possible reason for CW's superiority over NAPW/FAW is that an underbody CW mattress was always used with CW blankets, whereas no underbody blanket was used with FAW, which is the standard of care in our institution.

Previous studies have shown mixed effects of CW compared to FAW, likely due to variable device implementation, under-powered studies, and heterogeneity between studies, making it difficult to draw definitive conclusions. An extensive Cochrane library review that included 67 trials demonstrated a significant reduction in surgical site infections in patients who had FAW, although the quality of the evidence was found to be low [7]. The same review failed to demonstrate a significant reduction in blood transfusions in patients who were actively warmed, but did show benefit of prewarming in patients having major abdominal surgery. There was no difference in the effectiveness of any particular type of active warming system to another [7]. Our results may provide additional insight into heating methods, revealing a difference between CW and FAW without prewarming and thereby suggesting benefit to prewarming patients.

A systematic review from the Joanna Briggs Institute in Australia demonstrated that FAW was effective in maintaining intraoperative normothermia in comparison to no warming or passive warming, and that water garment warmers were significantly more effective than FAW in liver transplant patients [29]. This is generally consistent with our study, as water garments transfer heat similarly to CW.

Harper *et al.* reported that FAW was superior to CW; however, the CW intervention included only a CW underbody mattress, without a CW blanket on the anterior upper or lower body [18]. The study authors postulated that insensible losses may be higher amongst patients without anterior warming coverage in the CW arm, thereby increasing susceptibility to radiative and convective heat loss. Moreover, the primary endpoint of the study was single-timepoint core temperature at the conclusion of the case, rather than hypothermia throughout the surgery.

Other studies have shown equivalent benefit of CW compared to FAW but may have been underpowered (a total of four studies with 122 cumulative patients) [13–15, 17]. These investigations measured the duration of hypothermia rather than the incidence of post-operative hypothermia or single-timepoint measurement. A metanalysis of 5 studies concluded that FAW was superior to CW, and the study authors acknowledged the confounding nature of surface area coverage variability across arms, different temperature settings of heating devices, and different temperature measurement modalities [20]. Our study provides new data that may contradict some of these findings, notably that CW may be superior

to FAW when used without warming patients preoperatively (but with both an underbody CW blanket and covering CW blankets).

### Clinical implications

Our study suggests a benefit of CW vs NAPW/FAW and non-inferiority vs FAW/FAW, when an under-body CW heating mattress is used in combination with anterior upper or lower body coverage. It also supports the use of prewarming when using FAW alone without an underbody heated mattress. CW is reportedly 2.3 times more efficient compared to FAW in terms of energy use [12]. Moreover, CW blankets and mattresses are reusable, though they require sanitation with disposable wipes. CW produces less noise pollution compared to FAW [30]. Our study did not examine whether there could be a cost advantage to using non-disposable CW blankets compared to disposable FAW blankets; however, a cost advantage is not inconceivable, depending on the cost of acquisition and the life span of the product.

### Methodological considerations and limitations

We used AUC to measure the overall degree of hypothermia, which offers a more complete assessment of patient warming status than the incidence or duration of hypothermia alone. A majority of patients in our study did experience hypothermia (any temperature T<36°C). While the incidence was 74%, the duration of hypothermia was minimal and transient. The high incidence of hypothermia in this study may be due to factors including the sensitivity of the esophageal probe (to the tenths degree), enabling detection of hypothermia just 0.1°C below 36°C.

The study had the following limitations. Study enrollments fell short of the 216 minimum needed to detect a 1.0°C*hour difference in AUC with 80% power. The enrolled subjects were predominantly female (80%), reflecting the patient population presenting for abdominal, gynecologic, breast, plastic, and urological surgery in our hospital. Homogeneity by type of surgery was not perfect in the 10 randomization groups, as there were more patients in the abdominal and the gynecologic surgery groups than the other groups (breast, plastics/reconstructive, and urological) – see S1 Appendix – Randomization chart. Therefore, findings from the current study would benefit from confirmation in a larger clinical trial. Additionally, studies in individual surgical subtypes would likely be more representative of those individual populations.

Our study did not include CW/FAW and FAW/CW arms. Warmed cotton blankets were applied to patients in the NAPW arms in a non-standardized fashion, i.e., only patients who wanted them were covered with blankets, and the temperature of the blankets was not recorded. The intraoperative FAW arms did not have an underbody warming mattress or sheet, whereas the CW arms did. Lastly, the study was not powered for outcomes such as surgical site infections, amount of blood loss or blood transfusion, or the amount of post-operative shivering. Despite these shortcomings, the study provided clinically useful information about the ways in which to use active surface warming devices to maintain intraoperative normothermia.

### Conclusions

Our results suggest that CW is superior to FAW at reducing intraoperative hypothermia when FAW is used without prewarming. This refers to CW applied simultaneously as an underbody mattress and over-the-body cover. When patients can be actively prewarmed, CW/CW and FAW/FAW showed no difference in their ability to maintain normothermia. We conclude that CW may be routinely employed for perioperative warming both in the preoperative and intraoperative setting. This study was limited by not meeting its predefined sample size for enrollment.

### Supporting information

**S1 Appendix:  Randomization table.**
(XLSX)

**S2 Appendix: Mixed model for Randomized Block Design.**
(DOCX)

**S3 Appendix: Secondary outcomes and complications.**
(XLSX)

## Author contributions

**Conceptualization:** Ronak Desai, Ludmil Mitrev, Sandeep Krishnan, Kinjal Patel.

**Data curation:** Ronak Desai, Jason Gosschalk, Noud van Helmond, Ludmil Mitrev, Catherine Zhang, Brian McEniry, Krystal Hunter, Ernest Wallace, Michele Mele, Jennifer Ocbo, Keyur Trivedi, George Hsu, John DiBato, Kinjal Patel.

**Formal analysis:** Jason Gosschalk, Noud van Helmond, Krystal Hunter, John DiBato.

**Funding acquisition:** Ronak Desai, Noud van Helmond, Ludmil Mitrev.

**Investigation:** Ronak Desai, Jason Gosschalk, Noud van Helmond, Ludmil Mitrev, Catherine Zhang, Brian McEniry, Ernest Wallace, Michele Mele, Jennifer Ocbo, Keyur Trivedi, George Hsu, Kinjal Patel.

**Methodology:** Ronak Desai, Jason Gosschalk, Noud van Helmond, Ludmil Mitrev, Krystal Hunter, Michele Mele, Keyur Trivedi, Sandeep Krishnan, John DiBato.

**Project administration:** Ronak Desai, Jason Gosschalk, Noud van Helmond, Ludmil Mitrev, Catherine Zhang, Brian McEniry, Ernest Wallace, Michele Mele, Jennifer Ocbo, Keyur Trivedi, George Hsu.

**Resources:** Ronak Desai, Jason Gosschalk, Noud van Helmond, Ludmil Mitrev, Catherine Zhang, Brian McEniry, Krystal Hunter, Ernest Wallace, Michele Mele.

**Software:** Jason Gosschalk, Noud van Helmond, Krystal Hunter, John DiBato.

**Supervision:** Ronak Desai, Jason Gosschalk, Noud van Helmond, Ludmil Mitrev, Catherine Zhang, Brian McEniry, Krystal Hunter, Ernest Wallace, Michele Mele, Jennifer Ocbo, Keyur Trivedi, George Hsu, Kinjal Patel.

**Validation:** Ronak Desai, Jason Gosschalk, Noud van Helmond, Ludmil Mitrev, Catherine Zhang, Brian McEniry, Krystal Hunter, Ernest Wallace, Michele Mele, Keyur Trivedi, George Hsu, Sandeep Krishnan, John DiBato, Kinjal Patel.

**Visualization:** Jason Gosschalk, Noud van Helmond, Ludmil Mitrev, Krystal Hunter, John DiBato.

**Writing – original draft:** Ronak Desai, Jason Gosschalk, Noud van Helmond, Ludmil Mitrev, Catherine Zhang, Krystal Hunter, Ernest Wallace, Michele Mele, Jennifer Ocbo, Keyur Trivedi, George Hsu, Sandeep Krishnan, Kinjal Patel.

**Writing – review & editing:** Ronak Desai, Jason Gosschalk, Noud van Helmond, Ludmil Mitrev, Catherine Zhang, Brian McEniry, Krystal Hunter, Michele Mele, Jennifer Ocbo, Keyur Trivedi, George Hsu, Sandeep Krishnan, John DiBato, Kinjal Patel.

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
