## [Decision Letter · Decision Letter 0]

27 Nov 2024

Dear Dr. Desai,

Thank you for submitting your manuscript to PLOS ONE. After careful consideration, we feel that it has merit but does not fully meet PLOS ONE’s publication criteria as it currently stands. Therefore, we invite you to submit a revised version of the manuscript that addresses the points raised during the review process.

We look forward to receiving your revised manuscript.

Kind regards,

Amirmohammad Khalaji

Academic Editor

PLOS ONE

Journal Requirements: When submitting your revision, we need you to address these additional requirements. 1. Please ensure that your manuscript meets PLOS ONE's style requirements, including those for file naming. The PLOS ONE style templates can be found at https://journals.plos.org/plosone/s/file?id=wjVg/PLOSOne_formatting_sample_main_body.pdf and https://journals.plos.org/plosone/s/file?id=ba62/PLOSOne_formatting_sample_title_authors_affiliations.pdf

Reviewers' comments:

**Comments to the Author**

1. Is the manuscript technically sound, and do the data support the conclusions?

Reviewer #1: Yes

Reviewer #2: Yes

2. Has the statistical analysis been performed appropriately and rigorously?

Reviewer #1: Yes

Reviewer #2: Yes

3. Have the authors made all data underlying the findings in their manuscript fully available?

Reviewer #1: Yes

Reviewer #2: Yes

4. Is the manuscript presented in an intelligible fashion and written in standard English?

Reviewer #1: No

Reviewer #2: Yes

Reviewer #1: As a reviewer, I commend the authors for conducting a well-designed study to investigate the impact of different warming strategies on intraoperative hypothermia. The use of a randomized controlled trial and the objective measure of AUC below 36°C are the strengths of the study.

However, a few areas could be further explored:

Sample Size: Consider whether the sample size was adequate to detect clinically significant differences, especially for secondary outcomes, as it was mentioned in the study limitations.

Heterogeneity in Surgical Procedures: The inclusion of various surgical procedures might have influenced the results. A more homogeneous patient population could provide clearer insights.

Non-Standardized Blanket Use: The lack of standardization in blanket use in the NAPW arms is a limitation that could affect the results

The abstract is well-structured and concise. However, consider further condensing the methods section by focusing on the key aspects of the study design, such as the randomization process, intervention details, and primary outcome measure.

Revise the whole manuscript for any grammatical and language mistakes

Line 31: Consider rephrasing "We conducted a prospective observational study" to "We conducted a prospective randomized controlled trial."

Line 36: The equation for calculating AUC could be simplified or presented in a more concise format.

Line 48: Consider rephrasing "CW is associated with significantly less intraoperative hypothermia than FAW when FAW was used without prewarming" to "CW is more effective than FAW at reducing intraoperative hypothermia, especially when FAW is used without prewarming."

Line 54: Consider rephrasing "PH is associated with increased blood loss, increased time to emerge from anesthesia, prolonged hospitalization, surgical site infections, and general thermal discomfort" to "PH is associated with various adverse outcomes, including increased blood loss, prolonged emergence time, extended hospital stay, surgical site infections, and patient discomfort."

Line 61: The phrase "is there is no superiority" should be corrected to "there is no superiority."

Line 69: Consider rephrasing "Recently, conductive warming (CW) devices, which conduct heat from a thermal blanket directly to the patient, have become popular" to "Recently, conductive warming (CW) devices have gained popularity due to their ability to directly transfer heat to the patient."

Line 71: The phrase "CW devices have variable efficiency, but may be more efficient than FAW" could be rephrased to "While CW devices can vary in efficiency, they may offer superior heat transfer compared to FAW."

Line 73: Consider rephrasing "Studies comparing CW and FAW efficacy have been inconclusive" to "Studies comparing CW and FAW have yielded mixed results."

Line 272: Consider rephrasing "we assessed the association of four warming strategies" to "we compared the effectiveness of four warming strategies."

Line 276: Consider rephrasing "CW/CW was associated with 51% less hypothermia than NAPW/FAW" to "CW/CW resulted in 51% less hypothermia compared to NAPW/FAW."

Line 281: The phrase "No significant difference was observed in the pairwise comparisons between CW/CW and FAW/FAW, and NAPW/CW and FAW/FAW" is a bit convoluted. You could rephrase it as "No significant differences were observed between CW/CW and FAW/FAW, or between NAPW/CW and FAW/FAW."

Lines 289-291: Consider combining the two sentences to improve readability. For example, "One possible reason for CW's superiority over NAPW/FAW is the use of an underbody CW mattress, which was not used with FAW, despite being standard practice."

Line 293: The phrase "variable device implementation, low-powered studies, and heterogeneity between studies making conclusions difficult" could be rephrased for better clarity. For instance, "variable device implementation, underpowered studies, and heterogeneity between studies, making it difficult to draw definitive conclusions."

Lines 294-316: Consider breaking up the long paragraph into smaller paragraphs to improve readability. You could separate the discussion of the Cochrane review, the Joanna Briggs Institute review, and the specific studies by Harper et al. and others.

To improve the clarity and impact of your manuscript, please adhere to the following:

1. Prioritize clarity and conciseness in writing.

2. Ensure strong evidence base and clear methodology.

3. Address limitations and propose future directions.

4. Proofread carefully for grammar and style consistency.

Reviewer #2: This is a very interesting paper. This reviewer is participating in the re-submission. I have a few minor comments. Normally, when referring to the AUC of core body temperature, it suggests accumulated heat (i.e., heat storage), so it is somewhat difficult to interpret the AUC in the context of this experiment. Please add a figure with data from a representative subject to illustrate the AUC.

In the abstract, there is no explanation of T1 and T2. If possible, please state in the abstract that core body temperature during surgery was assessed using esophageal temperature. In the methods section, please provide the gender ratio of the subjects in each group. The impact of body size differences has been considered, but as a precaution, please present basic physical characteristics in a table. This should include age and gender ratio. I have no particular comments regarding the results and discussion, so please address the above points.

**Do you want your identity to be public for this peer review?** For information about this choice, including consent withdrawal, please see our Privacy Policy

Reviewer #1: **Yes: ** Hassan El-Masry

Reviewer #2: No

---

## [Author Response · Author response to Decision Letter 1]

9 Jan 2025

A detailed response to reviewers document has been uploaded with this revision

---

## [Decision Letter · Decision Letter 1]

27 Jan 2025

PONE-D-24-44335R1

The optimal warming strategy to reduce perioperative hypothermia: A prospective randomized non-blinded clinical trial

PLOS ONE

Dear Dr. Desai,

Thank you for submitting your manuscript to PLOS ONE. After careful consideration, we have decided that your manuscript does not meet our criteria for publication and must therefore be rejected.

I am sorry that we cannot be more positive on this occasion, but hope that you appreciate the reasons for this decision.

Kind regards,

Amirmohammad Khalaji

Academic Editor

PLOS ONE

Reviewers' comments:

Reviewer's Responses to Questions

**Comments to the Author**

Reviewer #1: All comments have been addressed

Reviewer #2: All comments have been addressed

2. Is the manuscript technically sound, and do the data support the conclusions?

Reviewer #1: Yes

Reviewer #2: Yes

3. Has the statistical analysis been performed appropriately and rigorously?

Reviewer #1: Yes

Reviewer #2: Yes

4. Have the authors made all data underlying the findings in their manuscript fully available?

Reviewer #1: Yes

Reviewer #2: Yes

5. Is the manuscript presented in an intelligible fashion and written in standard English?

Reviewer #1: No

Reviewer #2: (No Response)

**Reviewer #1:**  The discussion is so short with many passages without citations that strengthen their evidence

The methodology is deficient in many essential areas

**Reviewer #2: ** The authors responded to this reviewer's comments. However, the Y-axis label in Figure 1 should be the name of the temperature measured site, which is esophageal temperature.

**Do you want your identity to be public for this peer review?** For information about this choice, including consent withdrawal, please see our Privacy Policy

Reviewer #1: No

Reviewer #2: No

- - - - -

---

## [Author Response · Author response to Decision Letter 2]

24 Feb 2025

The Editors

PLOS One

RE: PONE-D-24-44335R1

Dear Sir/Madam:

Thank you for honoring our appeal and for your willingness to reconsider the manuscript. Below, you will find a point-by-point response to the questions from your editorial staff, as well as a full list of previous point-by-point memos from all prior revisions, in reverse chronological order. We are resubmitting the paper without changes to the body of the manuscript, since no specific need for changes was identified by the reviewers during the most recent review. We have only modified the labeling of the Y axis of Figure 1 to read “Esophageal temperature”. All requests for changes in R0 were addressed previously, documented in the chronology below. Methodological weaknesses have been disclosed in full in the manuscript. Our findings add value to perioperative care despite the shortcomings listed in the limitations section of the paper. This work concerns an under-researched area of perioperative care where little can be said with certainty about optimal patient warming methods, and this paper adds important insights to the small body of research that can help clinicians decide how best to warm up their patients.

Sincerely,

Ron Desai, DO

Associate Professor of Anesthesiology

---

## [Decision Letter · Decision Letter 2]

10 Apr 2025

Dear Professor Desai,

Thank you for submitting your manuscript to PLOS ONE. After careful consideration, we feel that it has merit but does not fully meet PLOS ONE’s publication criteria as it currently stands. Therefore, we invite you to submit a revised version of the manuscript that addresses the points raised during the review process.

We look forward to receiving your revised manuscript.

Kind regards,

Stefano Turi

Academic Editor

PLOS ONE

Journal Requirements:

Additional Editor Comments (if provided):

The authors should report in the description of Table 1 of the main manuscript how the data are presented in the table (values and values in brackets). 

Considering table 1, surgical approach is not similar in all groups; could the authors comment on this point?

Could the authors report the mean duration of surgery in each group? 

The planned sample size was not reached and this is an important limitation that could reduce the strength of conclusions. The authors should emphasize this point in the conclusion. 

Reviewers' comments:

Reviewer's Responses to Questions

**Comments to the Author**

Reviewer #3: All comments have been addressed

Reviewer #4: All comments have been addressed

2. Is the manuscript technically sound, and do the data support the conclusions?

Reviewer #3: Yes

Reviewer #4: Yes

3. Has the statistical analysis been performed appropriately and rigorously?

Reviewer #3: Yes

Reviewer #4: Yes

4. Have the authors made all data underlying the findings in their manuscript fully available?

Reviewer #3: Yes

Reviewer #4: Yes

5. Is the manuscript presented in an intelligible fashion and written in standard English?

Reviewer #3: Yes

Reviewer #4: Yes

Reviewer #3: This study evaluated the effectiveness of four different combinations of preoperative and intraoperative warming by comparing them to reduce perioperative hypothermia. The study design was rigorous, but some areas for improvement remain.

1.In the inclusion criteria, it is stated that the expected duration of the operation is more than 90 minutes and less than 240 minutes. This criterion is not supported by the literature, and it is hoped that the authors will add relevant evidence to enhance the rationality and scientific validity of the inclusion criteria.

2.The frequency of intraoperative temperature measurement using the esophageal probe placed after induction of anesthesia every 15 minutes was not supported in the literature. Since the frequency of temperature monitoring has a significant impact on the accuracy of the study results, it is recommended that the authors review and supplement the literature.

3.Considering that different types of surgeries may have different impacts on patients' temperature changes, has this study considered subgroup analyses of different types of surgeries to explore in more depth the effects of warming strategies under various types of surgeries?

Reviewer #4: (No Response)

**Do you want your identity to be public for this peer review?** For information about this choice, including consent withdrawal, please see our Privacy Policy

Reviewer #3: No

Reviewer #4: No

---

## [Author Response · Author response to Decision Letter 3]

8 May 2025

The authors should report in the description of Table 1 of the main manuscript how the data are presented in the table (values and values in brackets).

Author Responses

We apologize for this oversight and table 1 has been corrected.

Considering table 1, surgical approach is not similar in all groups; could the authors comment on this point?

Author Responses

Thank you for the comment. While some numerical differences existed in the proportions of patients who underwent open vs laparoscopic surgery across the four warming treatment arms, this difference was not found to be nominally significant (P > 0.05) on post-hoc testing of the distribution using a chi-squared test.

Could the authors report the mean duration of surgery in each group?

Author Responses

Mean duration of surgery has been added to Table 1.

The planned sample size was not reached and this is an important limitation that could reduce the strength of conclusions. The authors should emphasize this point in the conclusion.

Author Responses

Thank you for the suggestion, the Conclusion section on p19 has been modified to include a statement regarding this limitation (new text italicized):

“Our results suggest that CW is superior to FAW at reducing intraoperative hypothermia when FAW is used without prewarming. This refers to CW applied simultaneously as an underbody mattress and over-the-body cover. When patients can be actively prewarmed, CW/CW and FAW/FAW showed no difference in their ability to maintain normothermia. We conclude that CW may be routinely employed for perioperative warming both in the preoperative and intraoperative setting. This study was limited by not meeting its predefined sample size for enrollment.”

Reviewer #2:

Comments

Author Responses

GENERAL

This study evaluated the effectiveness of four different combinations of preoperative and intraoperative warming by comparing them to reduce perioperative hypothermia. The study design was rigorous, but some areas for improvement remain.

Author Responses

We thank the reviewer for their assessment.

SPECIFIC

In the inclusion criteria, it is stated that the expected duration of the operation is more than 90 minutes and less than 240 minutes. This criterion is not supported by the literature, and it is hoped that the authors will add relevant evidence to enhance the rationality and scientific validity of the inclusion criteria.

Author Responses

We thank the reviewer for their suggestion. However, when designing the study, we did not find a unified approach in terms of this inclusion criterion in the literature. For example, one previous randomized controlled trial included surgeries between 30-120 minutes in duration (Perl et al. Minerva Anesthesiol 2014;80:436-443), whereas another included surgeries 1-6 hours (Lau et al. Can J Anaesth 2018;65:1029-1040). We selected the expected duration of 90-240 minutes because, based on our clinical experience, it would allow for any hypothermia to occur while limiting the influence of any substantial outliers in very long surgical cases. The following was added to the Methods section on p6:

“Patients scheduled for surgery were screened for eligibility. Inclusion criteria were elective surgery under general anesthesia projected to last greater than 90 minutes but less than 240 minutes. The expected duration of 90-240 minutes was selected because, based on clinical experience, it would allow for any hypothermia to occur while limiting the influence of any substantial outliers in very long surgical cases.”

The frequency of intraoperative temperature measurement using the esophageal probe placed after induction of anesthesia every 15 minutes was not supported in the literature. Since the frequency of temperature monitoring has a significant impact on the accuracy of the study results, it is recommended that the authors review and supplement the literature.

Author Responses

In response, the following was added to the Methods section on p7:

“Consistent with a previous randomized controlled trial on warming strategies [24], patient temperature was measured (to the tenth degree Celsius) every 15 minutes intraoperatively using an esophageal probe placed after induction of anesthesia. IV fluids were not warmed. Room temperature was measured every 15 minutes using an Elitech RC-5 temperature logger (Elitech Technology Inc., San Jose, CA). Patients were contacted twice postoperatively, at 24 hours and then 30 days to assess for any complications not recorded in the electronic medical record.”

Considering that different types of surgeries may have different impacts on patients' temperature changes, has this study considered subgroup analyses of different types of surgeries to explore in more depth the effects of warming strategies under various types of surgeries?

Author Responses

We thank the reviewer for the suggestion. As described in the Methods section, stratified random sampling was employed to ensure equal distribution of surgery type across warming treatment arms. While this ensured an investigation in a sample of surgical patients broadly representative of varying surgical approaches in a large hospital, studies on individual surgery subtypes are likely more representative of those individual surgical populations. Subgroup analysis by surgery subtype in the present study would be inappropriate due to the limited sample size in individual surgical groups. To highlight this important point for readers the following point has been added to the Limitations section on p18:

“The study had the following limitations. Study enrollments fell short of the 216 minimum needed to detect a 1.0 °C*hour difference in AUC with 80% power. The enrolled subjects were predominantly female (80%), reflecting the patient population presenting for abdominal, gynecologic, breast, plastic, and urological surgery in our hospital. Homogeneity by type of surgery was not perfect in the 10 randomization groups, as there were more patients in the abdominal and the gynecologic surgery groups than the other groups (breast, plastics/reconstructive, and urological) – see S1 Appendix - Randomization chart. Therefore, findings from the current study would benefit from confirmation in a larger clinical trial. Additionally, studies in individual surgical subtypes would likely be more representative of those individual populations.”

---

## [Editor Report · Decision Letter 3]

22 May 2025

The optimal warming strategy to reduce perioperative hypothermia: A prospective randomized non-blinded clinical trial

PONE-D-24-44335R3

Dear Dr. Desai,

We’re pleased to inform you that your manuscript has been judged scientifically suitable for publication and will be formally accepted for publication once it meets all outstanding technical requirements.

Kind regards,

Stefano Turi

Academic Editor

PLOS ONE

---

## [Editor Report · Acceptance letter]

PONE-D-24-44335R3

PLOS ONE

Dear Dr. Desai,

I'm pleased to inform you that your manuscript has been deemed suitable for publication in PLOS ONE. Congratulations! Your manuscript is now being handed over to our production team.

Kind regards,

on behalf of

Dr. Stefano Turi

Academic Editor

PLOS ONE